# WaveFluid: A New Adversarial Approach for High-Fidelity Speech Synthesis

## Abstract

Probability flow based models for image and audio synthesis, such as denoising diffusion probabilistic models and Poisson flow generative models, can be interpreted as modeling the ground truth distribution through a non-compressible passive fluid partial differential equation, where the initial fluid density equals to ground truth distribution and the final fluid density equals to the chosen prior distribution. In this research, we improve the architectural designs of neural networks and propose WaveFluid model for mel-spectrogram conditioned speech synthesis task, which learns a velocity field directly through adversarial training instead of estimating the solution to a chosen linear partial differential equation like diffusion or Poisson equation in previous works. And since mel-spectrogram is a strong condition and limits the possible audios to a small range, we split our model into two stages and use reparameterization techniques to reduce memory footprint and improve training efficiency. Experimental results show that our model is competitive with previous vocoders in sample quality within five inference steps[1].

## 1 Introduction

Recent advancements in training algorithms and network architectures have facilitated the production of high-fidelity audio by deep generative models in the realm of speech synthesis (Kumar et al., 2019; Kong et al., 2020b; Lam et al., 2022; Huang et al., 2023; Lv et al., 2023; Ye et al., 2023). The pioneering implementation of a deep generative model involved the autoregressive generation of waveforms from mel-spectrograms (Oord et al., 2016; Kalchbrenner et al., 2018), which yielded high-fidelity audio but was hindered by a significantly slow inference speed. To overcome this limitation and achieve real-time high-fidelity speech synthesis, a multitude of non-autoregressive models have been proposed recently. These models can be broadly categorized into three types: flow-based models, generative adversarial networks, and diffusion probabilistic models.

Flow-based models generate waveforms from a chosen prior distribution, such as the Gaussian distribution, utilizing invertible neural networks (Ping et al., 2020; Prenger et al., 2019). These models require the preservation of invertibility and the evaluation of the determinant for training, which is accomplished through the employment of intricately designed neural networks. However, this design constrains the model's flexibility and restricts the quality of the audio output. In contrast, Generative Adversarial Networks (GANs) provide greater flexibility than flow-based models and can efficiently generate waveforms of superior fidelity (Kumar et al., 2019; Kong et al., 2020a; Kim et al., 2021). The success of these models can be attributed to the large receptive fields of the generators and the discriminators' capacity to identify noises of varying scales and periods. Specifically, Kumar et al. (2019) proposed multi-scale discriminators, while Kong et al. (2020a) introduced a multi-receptive field (MRF) generator and multi-period discriminators, significantly enhancing the model's performance.

Diffusion probabilistic models, which employ a Markov chain to transform a known prior distribution into a complex ground truth distribution, are the most popular choice (Kong et al., 2020b; Lam et al., 2022; Huang et al., 2023). These models utilize a noise-adding diffusion process without learnable parameters to obtain the training data for the denoising generator, eliminating the need

---

[1]Audio samples and codes are available at a newly registered anonymous repository: `https://github.com/JBJWZZHCDS/WaveFluild`

for additional networks such as discriminators or autodecoders during training. However, the inference process using diffusion models is typically time-consuming. To address this, Kong et al. (2020b), Lam et al. (2022), and Huang et al. (2023) have proposed several different approximate fast-sampling algorithms that can generate waveforms efficiently, albeit with a slight reduction in sample quality.

In this study, we initially conduct a review of diffusion probabilistic models and Poisson flow generative models, an efficient new visual generative model ((Xu et al., 2022)), under a unified perspective of a non-compressible passive fluid partial differential equation. This equation has boundary conditions at $t = 0$ and $t = 1$, which are equivalent to the ground truth distribution and a known prior distribution, respectively. These models employ a fixed linear partial differential equation (PDE), where the velocity field can be expressed as a function of fluid density. This PDE is then solved using the Green's function method [2] to derive the corresponding prior distribution and training data for the reverse denoising process.

Additionally, we improve the architectural designs of neural networks for generators and discriminators and introduce the WaveFluid model for the mel-spectrogram conditioned speech synthesis task. This model learns a velocity field directly through adversarial training. It is noteworthy that the mel-spectrogram, being a strong condition, restricts the potential audios to a narrow range. Consequently, we divide our model into two stages. The first stage is a deterministic function that upsamples mel-spectrograms to provide more detailed information for the second stage. The second stage, on the other hand, is a probabilistic refiner that uses velocity fields to generate high-fidelity waveforms based on the output of the first stage. We also employ reparameterization techniques in the second stage to minimize memory usage and enhance training efficiency. The mean opinion score (MOS) test results indicate that WaveFluid is on par with previous diffusion models and Generative Adversarial Networks (GANs) in terms of sample quality and efficiency.

## 2 BACKGROUNDS AND RELATED WORKS

### 2.1 NON-COMPRESSIBLE FLUID EQUATION

(1) shows the non-compressible fluid equation in physics.

$$\frac{\partial n}{\partial t}(\boldsymbol{x}, t) + \nabla_{\boldsymbol{x}}(n(\boldsymbol{x}, t)\boldsymbol{v}(\boldsymbol{x}, t)) - s(\boldsymbol{x}, t) = 0, \tag{1}$$

The fluid density function, $n(\boldsymbol{x}, t)$, the velocity field, $v(\boldsymbol{x}, t)$, and the source function, $s(\boldsymbol{x}, t)$, are vital when studying generative models. The primary concern in this case pertains to equations where the fluid density function, $n(\boldsymbol{x}, t)$, functions as a probability density function, represented as $p(\boldsymbol{x}, t)$. This perspective considers the initial boundary condition, $p(\boldsymbol{x}, 0) = p_{\text{data}}(\boldsymbol{x})$, where $p_{\text{data}}$ serves as the fundamental truth distribution from which samples can be obtained. Given that $\boldsymbol{v}(\boldsymbol{x}, t)$ is predetermined, it is possible for two distributions to transition into each other. For example, the final distribution at $t = 1$ may be derived from generating particles based on the data distribution $p(\boldsymbol{x}, 0) = p_{\text{data}}(\boldsymbol{x})$. The positions of these particles can subsequently be computed discretely through the application of a birth-death process (Lu et al., 2019), with $s(\boldsymbol{x}, t)$ dictating the probability of birth and death.

In contrast, should the chosen equation, which has previously reached a prior distribution at $t = 1$, be known, real data at $t = 0$ can be generated inversely by sampling from the prior distribution. Nevertheless, to maintain the density function as a probability density and facilitate efficient sampling without the birth-death process, the source, $s(\boldsymbol{x}, t)$, should be deemed as 0. With $s(\boldsymbol{x}, t) = 0$, a particle can initially be generated according to the known prior distribution. Subsequently, real data can be sampled from the trajectory Ordinary Differential Equation (ODE) from $t = 1$ to $t = 0$.

$$\frac{\mathrm{d}\boldsymbol{x}}{\mathrm{d}t} = \boldsymbol{v}(\boldsymbol{x}, t) \tag{2}$$

As (2) shows, the challenge now lies in identifying an appropriate velocity field, denoted as $\boldsymbol{v}(\boldsymbol{x}, t)$. Previous research has addressed this issue by solving specific linear partial differential equations (PDEs) with a single function variable, $\varphi(\boldsymbol{x}, t)$. In this context, $p(\boldsymbol{x}, t)$ and $\boldsymbol{v}(\boldsymbol{x}, t)$ can be considered

---

[2]Green's function method:https://en.wikipedia.org/wiki/Green%27s_function

as functions of $\varphi(\boldsymbol{x}, t)$. Utilizing the ground truth distribution and a Green's function solution to $\varphi(\boldsymbol{x}, t)$, along with the boundary condition at $t = 0$, the analytical forms of $p(\boldsymbol{x}, t)$ and $\boldsymbol{v}(\boldsymbol{x}, t)$ can be readily derived. Consequently, the velocity field can be trained efficiently. For example, the Gaussian perturbation kernels in diffusion probabilistic models can be interpreted as the Green's function to the diffusion equation. Detailed examples are provided in Appendix B and Appendix C.

## 2.2 SCORE BASED GENERATIVE MODELS

Models for audio synthesis, such as DiffWave, ProDiff (Huang et al., 2022b), and FastDiff, primarily focus on data scoring. Song et al. (2020) have successfully unified Noise Conditional Score Networks (Song & Ermon, 2019) and the Denoising Diffusion Probabilistic Model (Ho et al., 2020) under the umbrella of stochastic differential equations (SDEs). This unification is exemplified in the forward diffusion SDE, which is as (3).

$$\mathrm{d}\boldsymbol{x} = \boldsymbol{f}(\boldsymbol{x}, t)\mathrm{d}t + g(t)\mathrm{d}\boldsymbol{w} \tag{3}$$

And the corresponding backward SDE is shown in (4).

$$\mathrm{d}\boldsymbol{x} = [\boldsymbol{f}(\boldsymbol{x}, t) - g^2(t)\nabla_{\boldsymbol{x}} \log p(\boldsymbol{x}, t)]\mathrm{d}t + g(t)\mathrm{d}\boldsymbol{w} \tag{4}$$

The Kolmogorov forward equation, also known as the Fokker-Planck Equation[3] , can be reformulated into a linear fluid partial differential equation (5), and a detailed proof of this transformation is provided in Appendix A.

$$\frac{\partial p}{\partial t}(\boldsymbol{x}, t) + p(\boldsymbol{x}, t)\nabla_{\boldsymbol{x}}\boldsymbol{f}(\boldsymbol{x}, t) + \boldsymbol{f}(\boldsymbol{x}, t) \cdot \nabla_{\boldsymbol{x}}(p(\boldsymbol{x}, t)) - \frac{1}{2}g^2(t)\nabla_{\boldsymbol{x}}^2 p(\boldsymbol{x}, t) = 0, \tag{5}$$

$$\boldsymbol{v}(\boldsymbol{x}, t) = [\boldsymbol{f}(\boldsymbol{x}, t) - \frac{1}{2}g^2(t)\nabla_{\boldsymbol{x}} \log p(\boldsymbol{x}, t)]. \tag{6}$$

It's an non-compressible passive fluid linear partial differential equation where (6) exists, and the learning process of score based models is actually based on solving the equation by Green's function method. We will solve a special case $\frac{\partial p}{\partial t}(\boldsymbol{x}, t) - \nabla_{\boldsymbol{x}}^2 p(\boldsymbol{x}, t) = 0$ in Appendix B using Fourier Transformation to demonstrate this issue.

## 2.3 POISSON FLOW GENERATIVE MODELS (PFGMS)

Poisson Flow Generative Models (PFGMs) (Xu et al., 2022) are proficient visual generative models that exhibit comparable efficiency to score-based models. These models generate samples from the ground truth distribution by utilizing high-dimensional electric fields, which are solutions to the Poisson partial differential equation. To circumvent the issue of mode collapse, the original data is augmented with an additional dimension. The prior distribution is then defined as a uniform distribution on the surface of the superballs. It is noteworthy that this augmented dimension can be interpreted as time, thereby suggesting that PFGMs are essentially modeling a time-dependent Poisson equation as (7), where $\varphi(\boldsymbol{x}, t)$ is the electricity potential function.

$$\frac{\partial^2 \varphi}{\partial t^2}(\boldsymbol{x}, t) + \nabla_{\boldsymbol{x}}^2 \varphi(\boldsymbol{x}, t) = 0, \tag{7}$$

It is noteworthy that this equation cannot be directly interpreted as a fluid equation. To derive an appropriate equation, the selection (8) could be made.

$$p(\boldsymbol{x}, t) = \frac{\partial \varphi}{\partial t}(\boldsymbol{x}, t), \boldsymbol{v}(\boldsymbol{x}, t) = \frac{\nabla_{\boldsymbol{x}} \varphi(\boldsymbol{x}, t)}{\frac{\partial \varphi}{\partial t}(\boldsymbol{x}, t)}, \tag{8}$$

Thus, the boundary condition at $t = 0$ becomes (9).

$$p(\boldsymbol{x}, 0) = \frac{\partial \varphi}{\partial t}(\boldsymbol{x}, 0) = p_{\text{data}}(\boldsymbol{x}) \tag{9}$$

Presently, PFGMs are translated into a fluid equation, and training data can be generated subsequent to the resolution of this linear PDE. Further details regarding the Green's function solution to this equation, as well as the training process of PFGMs from the perspective of fluid equations, are deferred to Appendix C.

---

[3]Kolmogorov forward Equation or Fokker-Plank Equation: `https://en.wikipedia.org/wiki/Fokker%E2%80%93Planck_equation`

## 2.4 MELGAN AND HIFI-GAN VOCODER

As delineated in the introductory section, MelGAN proposes the use of multi-scale discriminators, while Hifi-GAN introduces the multi-receptive fusion generator and multi-period discriminators. These have subsequently become the foundational structures for current speech synthesis Generative Adversarial Networks (GANs). However, these models employ a single deterministic generator to directly produce waveforms from mel-spectrograms, while utilizing numerous discriminators to detect subtle noises. This approach, while efficient, inadvertently limits sample diversity and complicates further refinements, particularly when compared to diffusion probabilistic models. To address this issue, the subsequent section proposes novel model structures.

## 3 METHODS

### 3.1 OVERVIEW AND MOTIVATION

As previously discussed, our model can be viewed as a multi-step Generative Adversarial Network (GAN) that directly learns a velocity field through adversarial training and generates samples by discretely simulating the trajectory of an ordinary differential equation (ODE). However, this method is not universally efficient for all generative tasks. As the number of training steps increases, the stability of the training process decreases, and the memory footprint required for model training becomes prohibitively large.

Nevertheless, for the specific task of mel-conditioned speech synthesis, the potential audio outputs are already confined within a limited range. This implies that the primary audio information can be deterministically extracted from the provided mel-spectrograms, and probabilistic refinement should only be performed around the ground truth audio. In other words, from the perspective of the non-compressible fluid equation, a fluid particle, which represents an audio clip, is strongly attracted to a relatively small area where the real audio resides.

Guided by this fundamental task characteristic, we effectively developed a domain-specific, two-stage adversarial model for the mel-spectrograms conditioned speech synthesis task. In this model, the first generator corresponds to the deterministic upsample function, while the second refiner is accountable for the probabilistic refinement. It is noteworthy that these two networks are integrated into a single generator and trained concurrently. This implies that our model does not perform speech enhancement on a pre-trained GAN.

The general structures of WaveFluid are illustrated in Figure 1, and the detailed stuctures of modules can be found in Appendix D.

### 3.2 ARCHITECTURE

#### 3.2.1 GENERATOR

The generator is composed of two parts: (1) Attention Upsampler; (2) Velocity field refiner.

Attention Upsampler employs a mel-spectrogram as input, which is subsequently upsampled via an interpolation and upsample block. This process is detailed in the following paragraph. This procedure continues until the length of the output sequence aligns with the temporal resolution of the raw waveforms. Each output from the interpolation and upsampling process is aggregated and then fed into the subsequent layer. Our experimental results show that the upsampling shortcut is beneficial for decreasing the artifacts caused by transposed convolution.

The Velocity Field Refiner is predicated on the Unet model, utilizing the same upsample block as the generator and the same downsample block as the discriminator. The input for this component is derived from the output of the upsample blocks. Subsequently, the output is the average of the outputs from both the generator and the refiner.

#### 3.2.2 DWT BLOCK

In previous studies employing convolution-based networks, average pooling has been utilized to downsample raw audio. However, this approach overlooks the sampling theorem, resulting in the

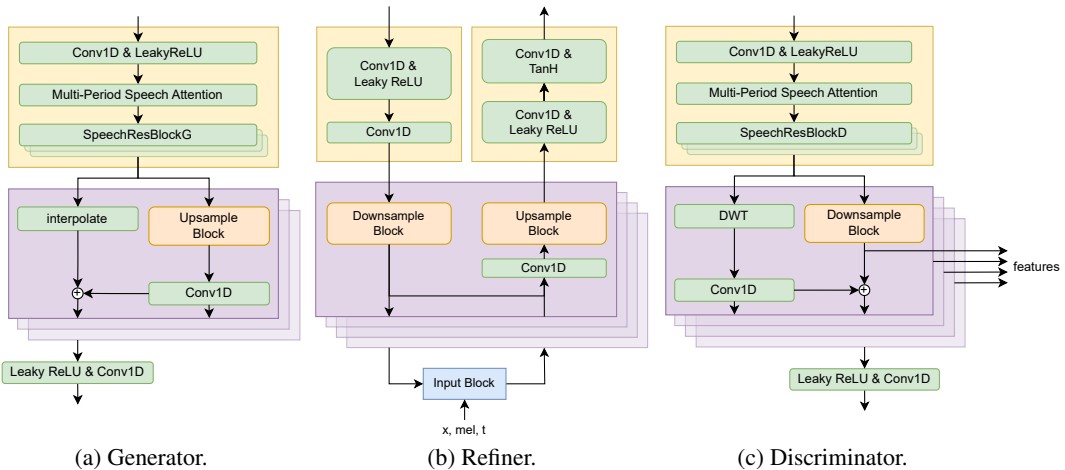

Figure 1: The structure of WaveFluid.

aliasing of high-frequency contents and rendering them invalid. In contrast,(Kim et al., 2021) Fre-GAN employs the Discrete Wavelet Transform (DWT) (Cohen et al., 1992) to execute non-destructive downsampling. We apply Daubechies1 wavelet transformation to our model, and this method preserves the high-frequency information of the waveform, making it more suitable for high-resolution audio processing. Detailed structure is visualized in Figure 4.

### 3.2.3 DISCRIMINATOR

The discriminator is symmetric with the first stage generator, but has quite different super parameters compared with the first stage generator. In addition, the upsampling shortcuts are changed into DWT shortcuts, and the upsample blocks are also replaced by downsample blocks, which have different residual blocks inside compared with the upsample blocks. The detailed module in discriminator can be seen in Figure 3.

### 3.2.4 ATTENTION BLOCKS

We have developed a specialized self attention (Vaswani et al., 2017) layer specifically tailored for periodic patterns inside audio data (detailed structure see Figure 5). Initially, the length of the input audio sequence is padded to become a multiple of the patching parameter, denoted as $p$. Subsequently, the sequence is divided into several groups, each containing a sequence of length $p$. Following the application of a 2D convolution layer with $3C_i$ output channels, the sequence is partitioned by the channel dimension into three distinct batches. Each batch is then reshaped to form $Q, K$, and $V$, where each patch rather than each sample point is regarded as a word and possesses an word vector with a dimension of $C_i p$.

$$Q, K, V = \text{conv2d}(\text{transpose}(\text{patch}(\text{input}))) \tag{10}$$

The conventional attention layer typically necessitates the multiplication of $Q$ and $K$, as (10) shows. However, for waveform data, the memory cost associated with this operation is prohibitively high. (Shen et al., 2021) proposed an innovative method to decouple $Q$ and $K$, and prioritize the multiplication of $K$ and $V$,which makes the algorithm's spatial complexity become $O(n)$, where $n$ is the length of audio sequence.The procedure is outlined as (11).

$$Q'_{i,j} = \frac{\exp(Q_{i,j})}{\sum_j \exp(Q_{i,j})}$$

$$K'_{i,j} = \frac{\exp(K_{i,j})}{\sum_i \exp(K_{i,j})} \tag{11}$$

$$\text{hidden} = Q'\left({K'}^\top V\right)$$

Finally, we restore the hidden state and adjust the sequence to match the original input shape as (12).

$$\text{out} = \text{unpad}(\text{unpatch}(\text{hidden})) \tag{12}$$

This design actualizes a self-attention mechanism while maintaining an acceptable balance between memory usage and computational cost. Finally, we adopt the dropkey method in (Li et al., 2023) to prevent overfitting.

## 3.3 Training Objective

**GAN Loss**   The training objectives for both the generator and the discriminator are derived from the LS-GAN(Mao et al., 2017) model, which substitutes the binary cross entropy components of the original GAN objectives with least square loss functions to ensure non-vanishing gradient flows. The discriminator is trained to classify authentic samples as 1, while samples synthesized from the generator are classified as 0. The generator, on the other hand, is trained to deceive the discriminator by enhancing the quality of the samples to be classified as nearly equivalent to 1. The losses for the generator (G) and the discriminator (D) within the GAN framework are defined as (13) and (14). Actually we also adopted WGAN(Arjovsky et al., 2017) training method,but only found similar performance with LS-GAN.

$$\mathcal{L}_{\text{Adv}}(D; G) = \mathbb{E}_{(x,s)}[(D(x) - 1)^2 + (D(G(s)))^2] \tag{13}$$

$$\mathcal{L}_{\text{Adv}}(G; D) = \mathbb{E}_s[(D(G(s)) - 1)^2] \tag{14}$$

**Reconstruction Loss**   In addition to the objectives of GAN, we incorporate mel-spectrogram loss and Discrete Wavelet Transform (DWT) loss. This integration is aimed at enhancing the training efficiency of the generator and improving the fidelity of the audio generated.

$$\mathcal{L}_{\text{Rec}}(G) = \mathbb{E}_{(x,s)} \left[ \lambda_{mel} \left\| \frac{\phi(x) - \phi(s)}{n} \right\|_1 + \lambda_{dwt} \left\| \frac{\varphi(x) - \varphi(s)}{n} \right\|_1 \right], \tag{15}$$

Reconstruction Loss is defined as (15), where $\phi$ is the function that transform a waveform into the corresponding mel-spectrogram, $\varphi$ is the composition of four DWT functions, and $n$ is the element count of $x$.

**Feature Matching Loss**   The feature matching loss is a learned similarity metric, quantified by the disparity in features of the discriminator between a ground truth sample and a generated sample. The definition of the feature matching loss is as (16).

$$\mathcal{L}_{\text{FM}}(G; D) = \mathbb{E}_{(x,s)} \left[ \sum_{i=1}^{T} \frac{1}{N_i} \left\| D^i(x) - D^i(G(s)) \right\|_1 \right] \tag{16}$$

**Final Loss**   Our final objectives for the GAN are listed in (17).

$$\begin{aligned} \mathcal{L}_{\text{G}} &= \lambda_{\text{adv}} \mathcal{L}_{\text{Adv}} + \lambda_{\text{fm}} \mathcal{L}_{\text{FM}}(G; D) + \mathcal{L}_{\text{Mel}}(G) \\ \mathcal{L}_{\text{D}} &= \mathcal{L}_{\text{Adv}}(D; G) \end{aligned} \tag{17}$$

Empirically, we set $\lambda_{\text{adv}} = 2$, $\lambda_{\text{fm}} = 15$, $\lambda_{\text{dwt}} = 8$ and $\lambda_{\text{mel}} = 50$ for the training process.

## 3.4 Training Algorithm

The training procedures for the proposed WaveFluid model are delineated as follows. Drawing inspiration from the aforementioned task principle, we employ the reparameterization technique to stabilize the training targets and enhance the training efficiency. Specifically, the model inputs mel-spectrograms, the outputs from the first stage, and the spatial-temporal coordinates into the second stage refiner. Instead of training the refiner to predict the velocity field $\boldsymbol{v}$, it is trained directly to predict the original data $x_{data}$. This approach allows us to obtain

$$\boldsymbol{v} = \frac{\boldsymbol{x}_{\text{predict}} - \boldsymbol{x}}{t}, \quad \forall\, t \in (0, 1]$$

through reparamterization process. The technique under discussion is specifically tailored for the domain of strong condition speech synthesis tasks. It possesses the potential to unify the training object for any arbitrary position, denoted as $x$, and time, denoted as $t$. This characteristic could substantially enhance the speed of convergence during the training process. The comprehensive training algorithm is delineated as follows:

---

**Algorithm 1:** Training WaveFluid

---

**Input:** first stage generator $G$, refiner $R$, discriminator $D$, mel condition $c$, time step $t$, time
decay rate range $[a, b]$

**begin**
   **repeat**
      Sample $\boldsymbol{x}_{\text{data}} \sim q_{\text{data}}(\boldsymbol{x}|c)$, $x \sim \mathbf{N}(0, \boldsymbol{I})$
      $t \leftarrow 1$
      $\boldsymbol{x}_{\text{hint}} \leftarrow G(c)$
      **for** $i = 1 \dots step - 1$ **do**
         $\Delta t \leftarrow t \times \text{uniform}(a, b)$
         $\boldsymbol{x}_{\text{predict}} \leftarrow R(\boldsymbol{x}_{\text{hint}}, t, \boldsymbol{x}, c)$
         $\boldsymbol{v} \leftarrow (\boldsymbol{x}_{\text{predict}} - \boldsymbol{x})/t$
         $\boldsymbol{x} \leftarrow \boldsymbol{x} + \boldsymbol{v} \times \Delta t$
         $t \leftarrow t - \Delta t$
      **end**
      $\boldsymbol{x} = R(\boldsymbol{x}_{\text{hint}}, t, \boldsymbol{x}, c)$
      $\boldsymbol{g} = (\boldsymbol{x} + \boldsymbol{x}_{\text{hint}})/2$
      Take gradient descent using common LSGAN with generated data $\boldsymbol{g}$ and real data $\boldsymbol{x}_{\text{data}}$
   **until** *WaveFluid converged*
**end**

---

As for the discrete inference schedule, due to the relatively strong robustness of our model, we could directly set all decay rate mentioned above to be $\frac{a+b}{2}$, which is the mean of training decay rate. And the scheduler could also be trained using gradient descent method if we fix the generator and the discriminator and optimize the time scheduler according to training loss.

## 4 EXPERIMENTS

### 4.1 EXPERIMENTAL SETUP

#### 4.1.1 DATASET

In order to ensure a fair and reproducible comparison against other competing methodologies, we utilize the LJSPeech dataset (Ito & Johnson, 2017), which comprises 13,100 audio clips of 22,050 Hz from a single female speaker, totaling approximately 24 hours of audio. To assess the model's generalization capabilities in multi-speaker scenarios, we employ the LibriSpeech ASR corpus, a large-scale corpus of read English speech amounting to 1,000 hours. We specifically select clear audio samples from this corpus and upsample these to 22,050Hz to align with the sampling rate of the LJSPeech dataset. Additionally, we utilize the Speech Command dataset for the Mean Opinion Score (MOS) (Ribeiro et al., 2011) test for unseen speakers. This dataset includes audio samples from human speakers in noisy environments. In accordance with standard practice, we conduct preprocessing and extract the spectrogram with a Fast Fourier Transform (FFT) size of 1024, a hop size of 256, and a window size of 1024.

#### 4.1.2 TRAINING AND EVALUATION

The detailed architectures and configurations of the models are listed in Appendix D. As for the traning process, the model is trained on a single Nvidia RTX 4090 GPU with a initial learning rate $2 \times 10^{-4}$ and a exponentially decay rate of $0.995$. The evaluation of audios' quality is conducted through 5-scale Mean Opinion Score (MOS) tests, which are crowd-sourced via Amazon Mechanical Turk. The MOS scores are documented with a 95% confidence interval. For the purpose of evaluation, each model generates 200 audio samples, half of which are derived from speeches by unseen speakers. Each sample is evaluated by two distinct workers. In addition to this, we employ supplementary objective evaluation metrics such as Short-Time Objective Intelligibility (STOI) (Taal et al., 2010) and Perceptual Evaluation of Speech Quality (PESQ) (Rix et al., 2001) to assess sample

equality. The real-time factor (RTF) assessment is also calculated, utilizing a single 3070Ti Laptop GPU.

## 4.2 COMPARSION WITH OTHER MODELS

We conduct a series of experiments on speech synthesis tasks to evaluate our model. Models we have compared with are listed below.

- **WaveGlow** (Prenger et al., 2019), an ancient parallel flow-based model;
- **WaveGrad** (Chen et al., 2020), **DiffWave** (Kong et al., 2020b), **BBDM** (Lam et al., 2022), and **FastDiff** (Huang et al., 2022a), four diffusion probabilistic models, all been proved to be high-fidelity. We use 50 denoising-steps for WaveGrad, 12 denoising-steps for BDDM, and 6 denoising-steps for DiffWave and FastDiff;
- **Hifi-GAN V1** (Kong et al., 2020a), a well-known GAN-based models;
- **WaveNet** (Oord et al., 2016), a autoregressive model.

We train these models following the setups as in the original papers,and the results in Table 1 show that our models is comparable with different kinds of previous models.

Table 1: Test results of different models on LJSpeech dataset.

| Model | MOS (↑) | STOI(↑) | PESQ(↑) | RTF (↓) |
|---|---|---|---|---|
| Ground Truth | $4.53 \pm 0.09$ | 1 | 4.598 | / |
| WaveNet(MOL) | $4.01 \pm 0.06$ | / | / | 307.6 |
| WaveGlow | $3.90 \pm 0.10$ | 0.950 | 3.20 | 0.062 |
| Hifi-GAN | $4.15 \pm 0.09$ | 0.947 | 3.57 | **0.018** |
| Diffwave (6 steps) | $4.18 \pm 0.11$ | 0.944 | 3.68 | 0.139 |
| WaveGrad (50 steps) | $4.04 \pm 0.05$ | 0.905 | 3.26 | 0.572 |
| FastDiff (4 steps) | $4.14 \pm 0.10$ | 0.953 | **3.72** | 0.044 |
| BDDM (12 steps) | $4.24 \pm 0.10$ | 0.955 | 3.66 | 0.267 |
| WaveFluid (1 step) | $4.19 \pm 0.11$ | **0.958** | 3.61 | 0.025 |
| WaveFluid (5 steps) | **$4.27 \pm 0.09$** | 0.951 | 3.64 | 0.058 |

Due to the adversarial training process, our model could generate relatively high quality audios with only one inference step, and the corresponding time consumption is significantly lower than usual diffusion models. What's more, different from traditional GAN models, our model could do further refinement which takes acceptable time to improve the sampling quality.

## 4.3 GENERALIZATION TO UNSEEN SPEAKERS

The generalizability of our proposed model is assessed utilizing two datasets: the LibriSpeech dataset and the SpeechCommands dataset. The Mean Opinion Score (MOS) is evaluated on the SpeechCommands dataset, which comprises a substantial amount of data collected in noisy environments. The Short-Time Objective Intelligibility (STOI) and Perceptual Evaluation of Speech Quality (PESQ) are examined on the LibriSpeech dataset, characterized by high-resolution audio data. The experimental outcomes for the melspectrogram inversion of the samples are delineated in Table 2. The results indicate that our model exhibits commendable performance in both high-noise and low-noise environments, exceeding the performance of the baseline models. And notably, it could consistently generates audio from speakers who were not included in the training set.

## 4.4 ABLATION STUDY

In order to demonstrate our structural designs are effective, we have conducted several ablation studies,and both subjective and objective evaluation results are presented in Table 3.

Our observations are concluded as follow:

Table 2: Test results of models to unseen speakers. It is worth mentioning that MOS was tested on Speech Command dataset, STOI and PESQ were tested on LibriSpeech ASR corpus.

| Model | MOS ($\uparrow$) | STOI($\uparrow$) | PESQ($\uparrow$) |
|---|---|---|---|
| Ground Truth | $4.29 \pm 0.08$ | 0.999 | 4.591 |
| WaveNet(MOL) | $3.85 \pm 0.12$ | / | / |
| WaveGlow | $3.82 \pm 0.09$ | 0.864 | 3.09 |
| Hifi-GAN | $4.04 \pm 0.09$ | 0.892 | 3.22 |
| Diffwave (6 steps) | $4.01 \pm 0.10$ | 0.890 | 3.19 |
| WaveGrad (50 steps) | $3.65 \pm 0.07$ | 0.848 | 3.02 |
| FastDiff (4 steps) | $4.02 \pm 0.13$ | 0.895 | 3.24 |
| BDDM (12 steps) | $4.08 \pm 0.11$ | 0.901 | **3.28** |
| WaveFluid (1 step) | $4.05 \pm 0.08$ | **0.907** | 3.23 |
| WaveFluid (5 steps) | **$4.11 \pm 0.12$** | 0.895 | 3.20 |

Table 3: Ablation study results.

| Model | MOS ($\uparrow$) | STOI($\uparrow$) | PESQ($\uparrow$) |
|---|---|---|---|
| Ground Truth | $4.53 \pm 0.09$ | 1 | 4.610 |
| WaveFluid (1 step) | $4.19 \pm 0.11$ | **0.958** | 3.61 |
| WaveFluid (5 steps) | **$4.27 \pm 0.09$** | 0.951 | 3.64 |
| w/o Period Attentions | $4.17 \pm 0.08$ | 0.946 | 3.55 |
| w/o Upsampling Shortcuts | $4.19 \pm 0.06$ | 0.948 | 3.60 |
| w/o DWT Shortcuts | $4.18 \pm 0.07$ | 0.945 | 3.56 |

1. The inference results of 1 step and 5 steps model show that our model, which contains a fluid equation based refiner, could improve the sample quality with a acceptable sacrifice in sampling speed.

2. The period attention blocks in the generator and discriminator is a effective module to capture long-range periodical patterns and improve sample quality.

3. The upsampling shortcuts in the generator and the DWT shortcuts in the discriminator are helpful to alleviate the artifacts caused by transposed convolution layers.

## 5 CONCLUSION

In conclusion, this study has provided a comprehensive review of diffusion probabilistic models and Poisson flow generative models, presenting them under a unified perspective of a non-compressible passive fluid partial differential equation. We have introduced the WaveFluid model, a novel approach to mel-spectrogram conditioned speech synthesis, which leverages enhanced architectural designs of neural networks for generators and discriminators. The model is divided into two stages: a deterministic function that upsamples mel-spectrograms and a probabilistic refiner that uses velocity fields to generate high-fidelity waveforms. The use of reparameterization techniques in the second stage has proven effective in minimizing memory usage and enhancing training efficiency.Finally, the results of the MOS test have demonstrated that the WaveFluid model is competitive with previous diffusion models and Generative Adversarial Networks (GANs) in terms of sample quality and efficiency.

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

# A    INTERPRETING SCORE BASED MODELS INTO LINEAR PDES

We consider the general Kolmogorov forward equation:

$$\mathrm{d}\boldsymbol{x} = \boldsymbol{\mu}(\boldsymbol{x},t)\mathrm{d}t + \boldsymbol{\sigma}(\boldsymbol{x},t)\mathrm{d}\boldsymbol{w}, \tag{18}$$

where $\boldsymbol{\mu}(\boldsymbol{x},t)$ is a vector function from $\mathbb{R}^n \times \mathbb{R}$ to $\mathbb{R}$, $\boldsymbol{\sigma}(\boldsymbol{x},t)$ is a matrix function from $\mathbb{R}^n \times \mathbb{R}$ to $\mathbb{R}^{n \times n}$, and $\mathrm{d}\boldsymbol{w}$ is the infinitesimal of n-dimentional standard Wiener process (also called Brown Motion)(Øksendal, 2003; 2013).

Now $\boldsymbol{x}(t)$ becomes a random variable, we denote its probability density function as $p(\boldsymbol{x},t)$. Assume $f$ is an arbitrary function $\in \mathcal{C}^{(2)}$, and $T$ is a arbitrary fixed positive time, using the tower property of conditioned expectation, we have:

$$\mathbb{E}[f(\boldsymbol{x}(T))] = \mathbb{E}[\mathbb{E}[f(\boldsymbol{x}(T))|\boldsymbol{x}(t) = \boldsymbol{x}]], \forall t \in [0,T], \tag{19}$$

we denote $\mathbb{E}[f(\boldsymbol{x}(T))|\boldsymbol{x}(t) = x]$ as $\boldsymbol{u}(x,t)$, then we have:

$$\mathbb{E}[f(\boldsymbol{x}(T))] = \int p(\boldsymbol{x},t)\mathbb{E}[f(\boldsymbol{x}(T))|\boldsymbol{x}(t) = x]\mathrm{d}x = \int p(\boldsymbol{x},t)u(\boldsymbol{x},t)\mathrm{d}x, \forall t \in [0,T], \tag{20}$$

then we denote the integration as a inner product between $p(\boldsymbol{x},t)$ and $u(\boldsymbol{x},t)$, and noticing that the left hand side has nothing to do with variable $t$, taking derivative at $t = T$ we have:

$$\begin{aligned}
0 = \left.\frac{\mathrm{d}\mathbb{E}[f(\boldsymbol{x}(T))]}{\mathrm{d}t}\right|_{t=T} &= \left.\frac{\mathrm{d}\langle p(\boldsymbol{x},t), u(\boldsymbol{x},t)\rangle}{\mathrm{d}t}\right|_{t=T} \\
&= \left\langle \left.\frac{\partial p(\boldsymbol{x},t)}{\partial t}\right|_{t=T}, u(\boldsymbol{x},T) \right\rangle + \left\langle p(\boldsymbol{x},T), \left.\frac{\partial u(\boldsymbol{x},t)}{\partial t}\right|_{t=T} \right\rangle,
\end{aligned} \tag{21}$$

now we obtain an equation with $\left.\frac{\partial p(\boldsymbol{x},t)}{\partial t}\right|_{t=T}$, where $\left.\frac{\partial u(\boldsymbol{x},t)}{\partial t}\right|_{t=T}$ could be further computed:

$$\begin{aligned}
\left.\frac{\partial u(\boldsymbol{x},t)}{\partial t}\right|_{t=T} &= \lim_{t\to 0^-} \frac{u(\boldsymbol{x},t+T) - u(\boldsymbol{x},T)}{t} \\
&= \lim_{t\to 0^-} \frac{\mathbb{E}[f(\boldsymbol{x}(T))|\boldsymbol{x}(t+T) = \boldsymbol{x}] - f(\boldsymbol{x})}{t},
\end{aligned} \tag{22}$$

then according to Itô [4] lemma we do Taylor expansion at $t = T$ for $f(x(t))$ and gain:

$$= -\left( \sum_{i=1}^n \mu_i(\boldsymbol{x},t)\left.\frac{\partial f(\boldsymbol{x})}{\partial x_i}\right|_{t=T} + \frac{1}{2}\sum_{i=1}^n\sum_{j=1}^n\sum_{k=1}^n \sigma_{ik}(\boldsymbol{x},t)\sigma_{jk}(\boldsymbol{x},t)\left.\frac{\partial^2 f(\boldsymbol{x})}{\partial x_i \partial x_j}\right|_{t=T} \right), \tag{23}$$

we regard this formula as a linear operator L act on function $f(\boldsymbol{x})$, where

$$\mathrm{L}(f)(\boldsymbol{x}) = \sum_{i=1}^n \mu_i(\boldsymbol{x},t)\left.\frac{\partial f(\boldsymbol{x})}{\partial x_i}\right|_{t=T} + \frac{1}{2}\sum_{i=1}^n\sum_{j=1}^n\sum_{k=1}^n \sigma_{ik}(\boldsymbol{x},t)\sigma_{jk}(\boldsymbol{x},t)\left.\frac{\partial^2 f(\boldsymbol{x})}{\partial x_i \partial x_j}\right|_{t=T} \tag{24}$$

$$\left.\frac{\partial u(\boldsymbol{x},t)}{\partial t}\right|_{t=T} = -\mathrm{L}(f)(\boldsymbol{x}) \tag{25}$$

Now we have transformed the SDE into equation:

$$\left\langle \left.\frac{\partial p(\boldsymbol{x},T)}{\partial t}\right|_{t=T}, f(\boldsymbol{x}) \right\rangle + \langle p(\boldsymbol{x},T), -\mathrm{L}(f)(\boldsymbol{x})\rangle = 0. \tag{26}$$

Since L is a linear operator, we could find its dual operator $\mathrm{L}^*$ with the integration inner product between functions using the formula of integration by parts: $(\langle \mathrm{L}(f), g\rangle = \langle f, \mathrm{L}^*g\rangle$ is the definition to dual operator $\mathrm{L}^*$)

$$\mathrm{L}^*(f)(x) = -\sum_{i=1}^n \left.\frac{\partial[\mu_i(\boldsymbol{x},t)f(\boldsymbol{x})]}{\partial x_i}\right|_{t=T} + \frac{1}{2}\sum_{i=1}^n\sum_{j=1}^n \left.\frac{\partial^2}{\partial x_i \partial x_j}\right|_{t=T} \left( \sum_{k=1}^n \sigma_{ik}(\boldsymbol{x},t)\sigma_{jk}(\boldsymbol{x},t)f(\boldsymbol{x}) \right), \tag{27}$$

---

[4]Itô lemma: https://en.wikipedia.org/wiki/It%C3%B4%27s_lemma

Now the SDE can be further transformed into:

$$\left\langle \left. \frac{\partial p(\boldsymbol{x}, t)}{\partial t} \right|_{t=T}, f(\boldsymbol{x}) \right\rangle - \langle \mathrm{L}^*(p)(\boldsymbol{x}, T), f(\boldsymbol{x}) \rangle = 0. \tag{28}$$

$$\left\langle \left. \frac{\partial p(\boldsymbol{x}, t)}{\partial t} \right|_{t=T} - \mathrm{L}^*(p)(\boldsymbol{x}, T), f(\boldsymbol{x}) \right\rangle = 0. \tag{29}$$

Since $f(\boldsymbol{x})$ is an arbitrary function $\in \mathcal{C}^{(2)}$, we have:

$$\left. \frac{\partial p(\boldsymbol{x}, t)}{\partial t} \right|_{t=T} - \mathrm{L}^*(p)(\boldsymbol{x}, T) = 0, \quad \forall\, T \in [0, +\infty), \tag{30}$$

$$\frac{\partial p(\boldsymbol{x}, t)}{\partial t} + \sum_{i=1}^{n} \frac{\partial [\mu_i(\boldsymbol{x}, t) p(\boldsymbol{x}, t)]}{\partial x_i} - \frac{1}{2} \sum_{i=1}^{n} \sum_{j=1}^{n} \frac{\partial^2}{\partial x_i \partial x_j} \left( \sum_{k=1}^{n} \sigma_{ik}(\boldsymbol{x}, t) \sigma_{jk}(\boldsymbol{x}, t) p(\boldsymbol{x}, t) \right) = 0, \tag{31}$$

and this is the partial equation that the probability density function should obey. Now we can review the simple situation:

$$\mathrm{d}\boldsymbol{x} = \boldsymbol{f}(\boldsymbol{x}, t)\mathrm{d}t + g(t)\mathrm{d}\boldsymbol{w} \tag{32}$$
$$\boldsymbol{\mu}(\boldsymbol{x}, t) = \boldsymbol{f}(\boldsymbol{x}, t), \boldsymbol{\sigma}(\boldsymbol{x}, t) = g(t)\boldsymbol{I}, \tag{33}$$

the equation can be simplified into:

$$\frac{\partial p(\boldsymbol{x}, t)}{\partial t} + \nabla_{\boldsymbol{x}}[\boldsymbol{f}(\boldsymbol{x}, t)p(\boldsymbol{x}, t)] - \frac{1}{2}g^2(t)\nabla_{\boldsymbol{x}}^2 p(\boldsymbol{x}, t) = 0, \tag{34}$$

$$\frac{\partial p(\boldsymbol{x}, t)}{\partial t} + p(\boldsymbol{x}, t)\nabla_{\boldsymbol{x}}\boldsymbol{f}(\boldsymbol{x}, t) + \boldsymbol{f}(\boldsymbol{x}, t) \cdot \nabla_{\boldsymbol{x}}p(\boldsymbol{x}, t) - \frac{1}{2}g^2(t)\nabla_{\boldsymbol{x}}^2 p(\boldsymbol{x}, t) = 0, \tag{35}$$

which is a linear non-compressible passive fluid partial differential equation.

## B    SOLVING STANDARD DIFFUSION EQUATION

Diffusion equation, which is also known as heat equation, is a parabolic partial differential equation that could be found in many PDE textbooks(Evans, 2022; John, 1991).

Firstly we derive the Green's function solution to the standard diffusion equation and we assume the source point $\boldsymbol{x}' = 0$ for simplicity:

$$\frac{\partial p(\boldsymbol{x}, t)}{\partial t} - \nabla_{\boldsymbol{x}}^2 p(\boldsymbol{x}, t) = \delta(\boldsymbol{x})\delta(t) , \tag{36}$$

the Fourier transformation of $p(\boldsymbol{x}, t)$ is denoted as:

$$\tilde{p}(\boldsymbol{k}, t) = \mathcal{F}[p] \equiv \int p(\boldsymbol{x}, t)e^{-i\boldsymbol{k}\cdot\boldsymbol{x}}\mathrm{d}^N\boldsymbol{x}, \tag{37}$$

the corresponding reverse Fourier transformation of $\tilde{p}(\boldsymbol{k}, t)$ is denoted as:

$$p(\boldsymbol{x}, t) = \mathcal{F}^{-1}[\tilde{p}] = \frac{1}{(2\pi)^N} \int \tilde{p}(\boldsymbol{k}, t)e^{i\boldsymbol{k}\cdot\boldsymbol{x}}d^N\boldsymbol{k}. \tag{38}$$

Fourier transformation's nice properties could remove the $\nabla_{\boldsymbol{x}}$ operator in some PDEs:

$$\mathcal{F}[\nabla_{\boldsymbol{x}}p] = i\boldsymbol{k}\tilde{p}, \mathcal{F}[\nabla_{\boldsymbol{x}}^2 p] = -|\boldsymbol{k}|^2\tilde{p}. \tag{39}$$

Apply Fourier transformation to the standard diffusion equation, we have:

$$\frac{\partial \tilde{p}}{\partial t} + |\boldsymbol{k}|^2\tilde{p} = \delta(t) ,$$

$$\Longleftrightarrow \qquad \frac{\partial \tilde{p}}{\partial t} + |\boldsymbol{k}|^2\tilde{p} = 0 \; (t > 0), \quad \tilde{p}(\boldsymbol{k}, 0) = 1 .$$

$$\Longleftrightarrow \qquad \tilde{p}(\boldsymbol{k}, t) = \exp(-|\boldsymbol{k}|^2 t), \tag{40}$$

which is a Gaussian distribution in $\boldsymbol{k}$ domain. Now we transform it back to $\boldsymbol{x}$ domain:

$$
\begin{aligned}
p(\boldsymbol{x}, t) = \mathcal{F}^{-1}[\tilde{p}] &= \frac{1}{(2\pi)^N} \int \exp(-|\boldsymbol{k}|^2 t)\exp(i\boldsymbol{k}\cdot\boldsymbol{x})\mathrm{d}^N\boldsymbol{k} \\
&= \prod_{j=1}^N \left[ \frac{1}{2\pi} \int_{-\infty}^{+\infty} \exp(ik_j x_j)\exp(-k_j^2 t)\mathrm{d}k_j \right] \\
&= \prod_{j=1}^N \left[ \frac{\exp(-\frac{x_j^2}{4t})}{2\pi} \int_{-\infty}^{+\infty} \exp\left[ -t\left(k_j - \frac{ix_j}{2t}\right)^2 \right]\mathrm{d}k_j \right] \\
&= \prod_{j=1}^N \left[ \sqrt{\frac{\pi}{t}} \cdot \frac{\exp\left(-\frac{x_j^2}{4t}\right)}{2\pi} \right] \\
&= \frac{1}{(4\pi t)^{\frac{N}{2}}} \exp\left( -\frac{|\boldsymbol{x}|^2}{4t} \right)
\end{aligned}
\tag{41}
$$

which is the Green's function solution whose source is at $\boldsymbol{x}' = \boldsymbol{0}$, thus for arbitrary source position:

$$
p(\boldsymbol{x}, t; \boldsymbol{x}') = \frac{1}{(4\pi t)^{\frac{N}{2}}} \exp\left( -\frac{|\boldsymbol{x} - \boldsymbol{x}'|^2}{4t} \right)
\tag{42}
$$

Now the diffusion equation could be solved by superposition method since the boundary condition at $t = 0$ could be regarded as $p_{\text{data}}(\boldsymbol{x})\delta(t)$:

$$
\begin{aligned}
p(\boldsymbol{x}, t) &= \int p(\boldsymbol{x}, t; \boldsymbol{x}')p_{\text{data}}(\boldsymbol{x}')\mathrm{d}^N\boldsymbol{x}' \\
\boldsymbol{v}(\boldsymbol{x}, t) = -\nabla_{\boldsymbol{x}}\log p(\boldsymbol{x}, t) &= -\frac{1}{p(\boldsymbol{x}, t)} \int \nabla_{\boldsymbol{x}} p(\boldsymbol{x}, t; \boldsymbol{x}')p_{\text{data}}(\boldsymbol{x}')\mathrm{d}^N\boldsymbol{x}' \\
&= \frac{1}{p(\boldsymbol{x}, t)} \int p(\boldsymbol{x}, t; \boldsymbol{x}')\frac{\boldsymbol{x} - \boldsymbol{x}'}{2t}p_{\text{data}}(\boldsymbol{x}')\mathrm{d}^N\boldsymbol{x}' \\
&= \int p(\boldsymbol{x}'|\boldsymbol{x}, t)\frac{\boldsymbol{x} - \boldsymbol{x}'}{2t}\mathrm{d}^N\boldsymbol{x}' \\
&= \mathbb{E}_{x\sim p(\boldsymbol{x}'|\boldsymbol{x}, t)}\left[ \frac{\boldsymbol{x} - \boldsymbol{x}'}{2t} \right]
\end{aligned}
\tag{43}
$$

where

$$
\begin{aligned}
p(\boldsymbol{x}'|\boldsymbol{x}, t) &\propto p_{\text{data}}(\boldsymbol{x}')p(\boldsymbol{x}, t; \boldsymbol{x}') \\
&\propto p_{\text{data}}(\boldsymbol{x})\exp\left( -\frac{|\boldsymbol{x} - \boldsymbol{x}'|^2}{4t} \right)
\end{aligned}
\tag{44}
$$

when $t$ is large enough, $p(\boldsymbol{x}, t)$ is approximately proportional to $\exp\left(-\frac{|\boldsymbol{x}-\boldsymbol{x}'|^2}{4t}\right)$, which could serve as a prior distribution. Now the inference process has a proper beginning and the velocity field could be trained efficiently through adding Gaussian noises to the origin clear data like diffusion probabilistic models. It's worth mentioning that, this conditioned expectation is also similar to another efficient training objective for diffusion models called stable target field objective (Xu et al., 2023), which means that the original data could be regarded as point charges, Green's function determine the analytical form of the electricity field, and the velocity field could be viewed as the join electricity field of the point charges.

The process of inference now commences appropriately, and the velocity field can be effectively trained by incorporating Gaussian noises into the original, unadulterated data, akin to diffusion probabilistic models. It is noteworthy that this conditioned expectation bears resemblance to another efficient training objective for diffusion models, referred to as the stable target field objective (Xu et al., 2023). This implies that the original data can be conceptualized as point charges, with Green's function determining the analytical form of the electric field. Consequently, the velocity field can be perceived as the combined electric field of the point charges.

## C  SOLVING TIME-DEPENDENT POISSON EQUATION

Firstly,we also needs to fine the Green's function solution:

$$\frac{\partial^2 \varphi}{\partial t^2}(\boldsymbol{x}, t) + \nabla_{\boldsymbol{x}}^2 \varphi(\boldsymbol{x}, t) = \delta(\boldsymbol{x})\delta(t)$$

$$\Longleftrightarrow \quad \frac{\partial^2 \varphi}{\partial t^2}(\boldsymbol{x}, t) + \nabla_{\boldsymbol{x}}^2 \varphi(\boldsymbol{x}, t) = 0(t > 0), \quad \frac{\partial \varphi}{\partial t}(\boldsymbol{x}, 0) = \delta(\boldsymbol{x}) \tag{45}$$

Similar to Appendix B, we apply Fourier transformation to the equation:

$$\frac{\partial^2 \tilde{\varphi}}{\partial t^2}(\boldsymbol{k}, t) - |\boldsymbol{k}|^2 \tilde{\varphi}(\boldsymbol{k}, t) = 0(t > 0), \quad \frac{\partial \tilde{\varphi}}{\partial t}(\boldsymbol{x}, 0) = \delta(x)$$

$$\Longleftrightarrow \quad \tilde{\varphi}(\boldsymbol{k}, t) = \frac{u \exp(-|\boldsymbol{k}|t) + v \exp(|\boldsymbol{k}|t)}{|\boldsymbol{k}|}, \quad -u + v = 1 \tag{46}$$

Since $t \to \infty, \varphi(\tilde{\boldsymbol{k}}, t) \to 0$,we have $u = -1, b = 0, \tilde{\varphi}(\boldsymbol{k}, t) = \frac{1}{|\boldsymbol{k}|} \exp(-|\boldsymbol{k}|t)$,then apply reverse Fourier transformation with some properties of hypergeometric[5]function and n-dimensional spherical coordinates mentioned in (Liu et al., 2023):

$$\varphi(\boldsymbol{x}, t) = \frac{\Gamma\left(\frac{N-1}{2}\right)}{2\pi^{\frac{N+1}{2}}} \frac{1}{(t^2 + |\boldsymbol{x}|^2)^{\frac{N-1}{2}}}, \tag{47}$$

which is the n-dimensional electricity potential function of a unit point charge at $\boldsymbol{x}' = 0$, and for arbitraty source position $\boldsymbol{x}$,we have:

$$\varphi(\boldsymbol{x}, t; \boldsymbol{x}') = \frac{\Gamma\left(\frac{N-1}{2}\right)}{2\pi^{\frac{N+1}{2}}} \frac{1}{(t^2 + |\boldsymbol{x} - \boldsymbol{x}'|^2)^{\frac{N-1}{2}}}, \tag{48}$$

Actually since Poisson equation is very special, a more simpler method to solve it could be found in PFGMs' original paper(Xu et al., 2022). Now we have:

$$p(\boldsymbol{x}, t) = \frac{\partial \varphi}{\partial t}(\boldsymbol{x}, t) = \int \frac{\partial \varphi(\boldsymbol{x}, t; \boldsymbol{x}')}{\partial t} p_{\text{data}}(\boldsymbol{x}') \mathrm{d}\boldsymbol{x}' \tag{49}$$

$$\boldsymbol{v}(\boldsymbol{x}, t) = \frac{\nabla_{\boldsymbol{x}} \varphi(\boldsymbol{x}, t)}{\frac{\partial \varphi}{\partial t}(\boldsymbol{x}, t)} = \frac{1}{p(\boldsymbol{x}, t)} \int \nabla_{\boldsymbol{x}} \varphi(\boldsymbol{x}, t; \boldsymbol{x}') p_{\text{data}}(\boldsymbol{x}') \mathrm{d}^N \boldsymbol{x}',$$

$$= \frac{1}{p(\boldsymbol{x}, t)} \int \frac{\partial \varphi(\boldsymbol{x}, t; \boldsymbol{x}')}{\partial t} \frac{\boldsymbol{x} - \boldsymbol{x}'}{t} p_{\text{data}}(\boldsymbol{x}') \mathrm{d}\boldsymbol{x}'$$

$$= \mathbb{E}_{x \sim p(\boldsymbol{x}'|\boldsymbol{x}, t)} \left[ \frac{\boldsymbol{x} - \boldsymbol{x}'}{t} \right] \tag{50}$$

where

$$p(\boldsymbol{x}'|\boldsymbol{x}, t) \propto p_{\text{data}}(\boldsymbol{x}') \frac{\partial \varphi(\boldsymbol{x}, t; \boldsymbol{x}')}{\partial t}$$

$$\propto \frac{p_{\text{data}}(\boldsymbol{x}')}{(t^2 + |\boldsymbol{x} - \boldsymbol{x}'|^2)^{\frac{N+1}{2}}} \tag{51}$$

Then we could use a training process that is very similar to diffusion models in Appendix B to train this velocity field by changing Gaussian perturbation kernel according to corresponding Green's function.

## D  MODEL STRUCTURE DETAILS

---

[5]hypergeometric function:https://en.wikipedia.org/wiki/Hypergeometric_function

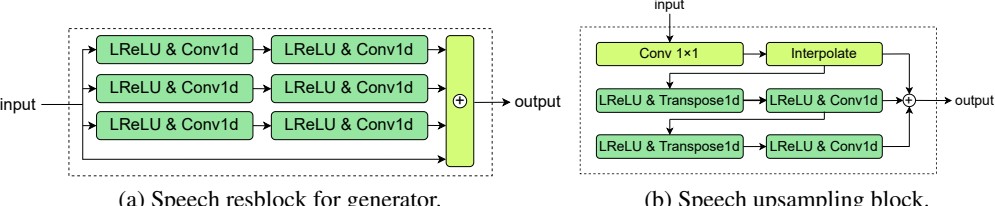

Figure 2: Modules in generator block.

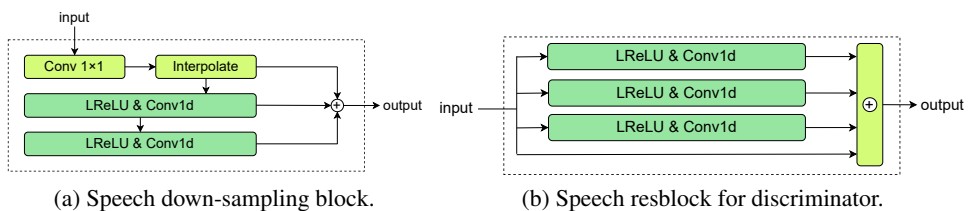

Figure 3: Modules in discriminator block.

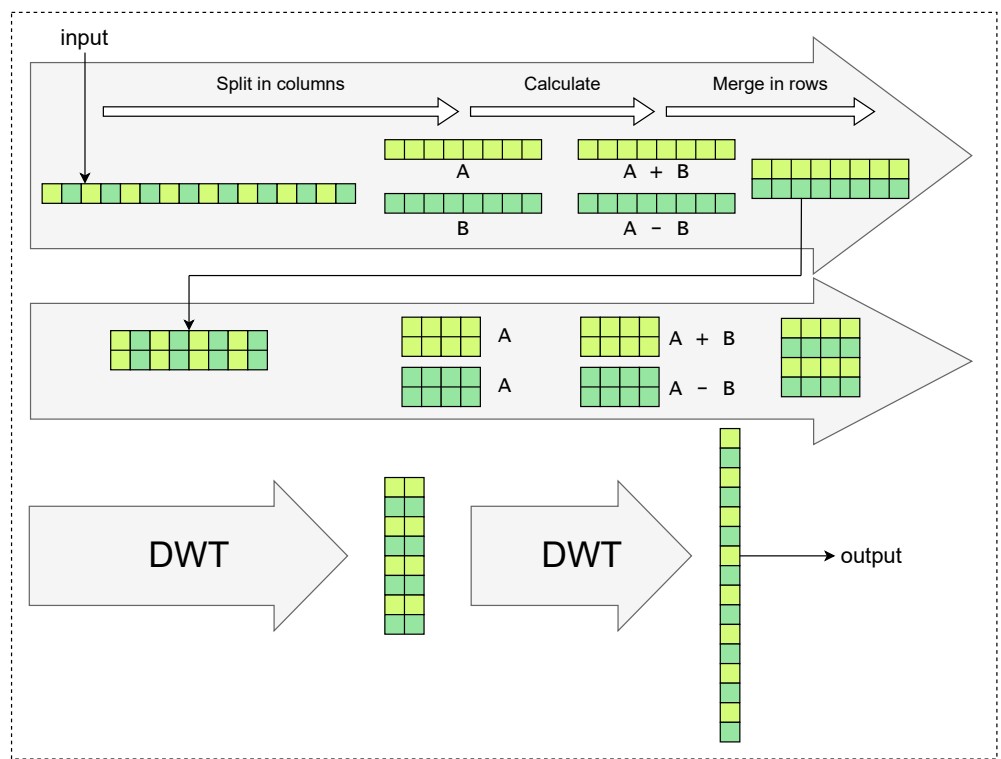

Figure 4: The block of Discrete Wavelet Tranformation(DWT).

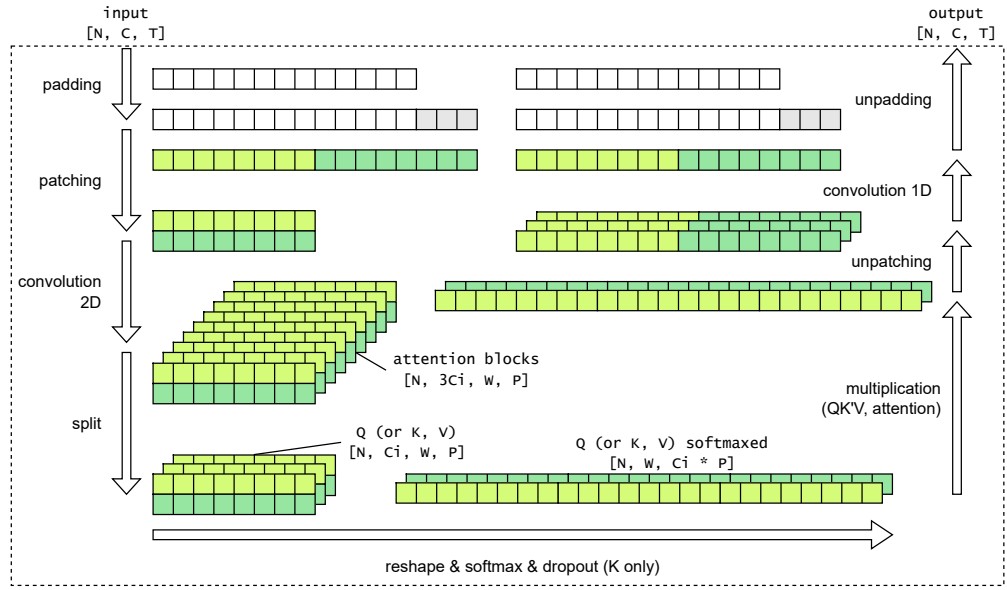

(a) A single speech attention block.

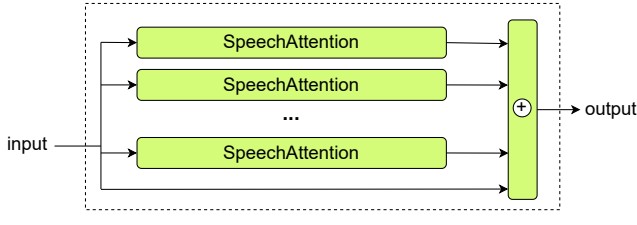

(b) Multi-period speech attention block.

Figure 5: Attention Blocks in Wavefluid.

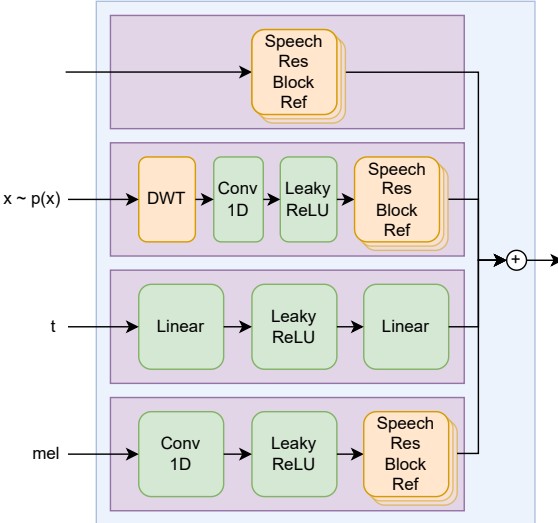

Figure 6: Input block in Refiner.

Table 4: Model params.

| Param | Value |
|---|---|
| useWeightNorm | True |
| AdamW Betas&WeightDecay | [ betas=(0.8,0.99), weightDecay=0.01] |
| channelsGen | [256,128,64,64] |
| attGroupsGen | [1,1,1,1] |
| patchesGen | [[2,3],[2,3],[2,3,5],[2,3,5,7]] |
| attMidChannelsGen | [64,32,16,16] |
| dropRateGen | [0.3,0.25,0.2,0.15] |
| resGroupsGen | [1,1,1,1] |
| resMidChannelsGen | [256,128,64,64] |
| upSampleRates | [8,8,4] |
| upGroups | [1,1,1] |
| downSampleRatesRefiner | [8,4,4,2] |
| channelsRefiner | [32,64,128,256,256] |
| groupsRefiner | [4,8,16,32] |
| channelsDis | [64,128,256,512,1024] |
| attGroupsDis | [4,8,16,32,64] |
| patchesDis | [[2,3,5,7,11],[2,3,5,7],[2,3,5,7],[2,3,5],[2,3,5]] |
| attMidChannelsDis | [32,64,128,256,512] |
| dropRateDis | [0.3,0.25,0.2,0.15,0.1] |
| resGroupsDis | [4,8,16,32,64] |
| resMidChannelsDis | [64,128,256,512,1024] |
| downSampleRates | [2,2,4,4] |
| downGroups | [8,16,32,64] |

