# OpenReview forum: "WaveFluid: A New Adversarial Approach for Efficient High-Fidelity Speech Synthesis"
_ICLR.cc/2024/Conference — ICLR 2024 Conference Withdrawn Submission_

### Official Review · Reviewer_D9ch · 2023-10-28

**Soundness:** 2 fair
**Presentation:** 2 fair
**Contribution:** 2 fair
**Rating:** 3
**Confidence:** 5

**Summary:**

They proposed a novel neural vocoder with an efficient iterative structure. They introduced new generator and discriminator architectures for waveform generation, and integrated a waveform generation and speech enhancement task in a single pipeline.

**Strengths:**

This work utilizes DWT to extract efficient and fidelity high-frequency features of audio for high-resolution waveform audio generation.

They trained the generator and refiner which is like a speech enhancement model. However, two structures are jointly trained for high-quality waveform generation.

**Weaknesses:**

Although this work generated a good performance in single dataset, I think it is not enough to demonstrate the practicality of this model. Specifically, LJSpeech is very easy to reconstruct with any model. I suggest some further experiments for your work.

1. It would be better to train the model with a multi-speaker dataset such as LibriTTS, and compare it with BigVGAN [ICLR 2023] and Vocos.

2. It is also important to verify that this model has good performance within the TTS pipeline so the TTS results with the same Mel-spectrogram generation model should be included. Please refer HiFi-GAN paper for the TTS experiment.

3. I know that it is difficult to evaluate audio quality so I hope to utilize much more metrics to evaluate your model such as Pitch, Periodity, and Voice/Unvoice F1 Score which were proposed in CARGAN [ICLR 2022].

4. It would be useful to compare different combinations of discriminators such as MPD, MSD, MRD, and MS-STFTD. In my experience, a multi-period discriminator is important to generate a perceptually high-quality waveform. It would be useful to compare different combinations of discriminators such as MPD, MSD, MRD, and MS-STFTD. In my experience, a multi-period discriminator is important to generate a perceptually high-quality waveform.

For DWT, Fre-GAN also utilized it to preserve the high-frequency information of the waveform not even in downsampling. It seems that both discriminators have almost the same architecture.

**Questions:**

1. I have a question about sample diversity in neural vocoder. In section 2.4, you described that previous GAN-based neural vocoders have a limitation for sample diversity. However, I think neural vocoder do not need sample diversity, but they may need fidelity. They only reconstruct waveform from Mel-spectrogram. It is slightly different from the diversity of current generative AI models. Could you explain in more detail for sample diversity you think and how to evaluate the sample diversity in your paper. I could not see this part. Furthermore, if you said about out-of-distribution generation, you should have trained your model with a multi-speaker dataset and evaluated it with an extremely noisy dataset.

2. Have you tried to train the generator and refiner respectively? If you trained your refiner separately, refiner would be utilized for any vocoders, right?

3. Where is the demo page for audio samples? In my very personal opinion, it is mandatory for speech paper to create a demo page.

---

### Official Review · Reviewer_WXsx · 2023-10-28

**Soundness:** 2 fair
**Presentation:** 3 good
**Contribution:** 2 fair
**Rating:** 5
**Confidence:** 4

**Summary:**

This work proposes the WaveFluid model, a novel vocoder that incorporates improved architectural designs for DiffGAN-based systems. The model consists of two stages: a deterministic function that upsamples mel-spectrograms and a probabilistic refiner that utilizes velocity fields to generate high-fidelity waveforms. The experimental results demonstrate the superior performance of the WaveFluid model in terms of sample quality and efficiency.

**Strengths:**

1. The proposed two-stage generator that includes a deterministic function that upsamples mel-spectrograms and a probabilistic refiner that uses velocity fields to generate high-fidelity waveform is interesting and effective.
2. The effectiveness of the proposed period attention blocks has been well proven by the ablation studies.
3. The upsampling shortcuts in the generator and the DWT shortcuts in the discriminator are helpful to alleviate the artifacts caused by transposed convolution layers.
4. The proposed fluid equation based refiner could improve the sample quality with a acceptable sacrifice in sampling speed.

**Weaknesses:**

1. Although the proposed architectural designs of WaveFluid are interesting, my main concerns are about the novelty of the theoretical innovation. In Section 1, the authors claim that "We also employ reparameterization techniques in the second stage to minimize memory usage and enhance training efficiency" and "Instead of training the refiner to predict the velocity field v, it is trained directly to predict the original data". Although the authors present diffusion probabilistic models and Poisson flow generative models under a unified perspective of a non-compressible passive fluid partial differential equation, the WaveFluid model seems to be a DiffGAN-based model [1] in terms of the prediction target defined in Section 3.3. Besides, the authors should make comparisons about whether to predict the velocity field v or not using the same model architecture to support their claims about memory usage and training efficiency.
2. In Table 3, whether the result of "w/o Period Attentions" is derived from the "WaveFluid (1 step)" or "WaveFluid (5 steps)" is not very clear. Adding some relevant explanations here may enhance the clarity of the article.
3. In the abstract, the authors claim that "we split our model into two stages and use reparameterization techniques to reduce memory footprint and improve training efficiency". However, in Section 4, the experimental results of the memory footprint and training efficiency are missing.
4. In Table 3, the evaluations of RTF and memory footprint are missing. The computational costs of the proposed Period Attentions modules should be included.
5. Typos:
In Section 3.2.4, Line 13, "K and V,which" -> "K and V, which";
In Section 3.2.4, Line 14, "audio sequence.The procedure is" -> "audio sequence. The procedure is";
In Section 5, Line 8, "training efficiency.Finally" -> "training efficiency. Finally";
In Appendix A, Line 6, "p(x, t).Assume" -> "p(x, t). Assume";


I would like to raise my scores if the authors have fully addressed my concerns.


[1] Huang, Rongjie, et al. "FastDiff 2: Revisiting and Incorporating GANs and Diffusion Models in High-Fidelity Speech Synthesis." Findings of the Association for Computational Linguistics: ACL 2023. 2023.

**Questions:**

My main concerns and questions are mainly related to the theoretical novelty and experiments. Please refer to the above Weaknesses section for details.

---

### Official Review · Reviewer_kxZL · 2023-11-06

**Soundness:** 3 good
**Presentation:** 1 poor
**Contribution:** 3 good
**Rating:** 6
**Confidence:** 3

**Summary:**

The authors propose a few things:

1. A unified mathematical framework for understanding DDPMs, flow-based models, and Poisson flow networks.
2. They propose a new architecture for spectrogram vocoding in a 2-stage pipeline (trained jointly). They evaluate this against existing vocoders and find similar-to-slightly-improved performance.
3. To train their GAN vocoder, they introduce a new iterative refinement method for GANs. A noisy waveform is iteratively refined into a clean waveform, whereby at each iteration they linearly interpolate the running prediction toward the GAN model output with some weight for a fixed number of steps (1-5 typically). Some theory is given around how this iterative linear interpolation between the model output and running prediction (starting from noise) relates to non-compressible fluid equations.

**Strengths:**

There are 2 main strengths of the paper:

(1)
The idea of iterative refinement of the predictions of a GAN model is a really cool idea, and I am not surprised to see it yield improved results. Perhaps it was necessary to view it from the perspective of DDPMs and velocity fields to realize that the simple idea of linearly interpolating a running prediction toward GAN predictions could be so handy. Very solid, original, core idea.

(2)
The construction of the speech attention block is nice and quite clever. The diagram for it indicated shapes and made understanding it quite clear. The ablations demonstrated that this component is important to the model's performance, and given its construction, I can understand why. Good idea, and well executed! Can probably be widely used if made easy to integrate into new models.

There is a third, lesser strength of the paper: the attempt to pull score based models, Poisson flow models, and DDPMs under a single, physics-based framework, is an original and interesting idea. It was perhaps not communicated so successfully, but certain parts of the discussion around it were engaging and persuasive.

**Weaknesses:**

Starting with the least significant and progressing to the most significant weaknesses:

# Typos and grammar

- Equation (1) has 'v' bolded, but in the text below it is not bolded. Please clarify if 'v' here is a vector field or a scalar field? i.e. is $v(\mathbf{s}, t)$ a vector or a scalar?
- Equation (3) introduces $g(t)$, but $g(t)$ is never defined/explained except implicitly through the citation. Please define what $g(t)$ represents in the manuscript.
- Page 14 near the bottom, same few sentences repeated twice.
- Appendix C, first sentence grammar check. "we also needs to fine" --> "we need to find". Also missing spaces after commas in third sentence on the page. Also after the footnote mark.
- The paper repeatedly switches between citing without a space (e.g. " LS-GAN(Mao et al., 2017)" ), and with a space (e.g. "process (Lu et al., 2019)"). Please make it consistent.
- Lots of strange spacing in the text of the paper. Many missing (or too many) spaces after full-stops and other punctuation. Please remedy in final manuscript.
- References: LJSpeech dataset capitalization.
- References: HiFi-GAN capitalization
- References: Several more capitalization typos, please double check proper nouns are capitalized appropriately.
- Section 3.2.1, "This process is detailed in the following paragraph" --> not detailed in the next paragraph.
- Section 3.2.3, "quite different super parameters" -- what is meant by this? I am not aware of 'super parameters' being a widely used term? Maybe you mean hyper-parameters?
- "an word vector", and a few other grammar mistakes.
- Equation (10) does not show that Q must be multiplied with K, unfortunately. Rather just remove the clause "as (10) shows.", since it just introduces confusion.
- Section 3.3, first paragraph has inconsistent tense. Last sentence has very bad grammar.
- Several other minor grammar, tense, wording, or copy-paste typos that make the content harder to understand in several areas.

# Significant weaknesses

Now, on to the more significant weaknesses, which there are primarily 2:

1. Subjective evaluations: Only 2 speakers evaluating over each model is not enough to be robust. Much work has been done on this, and -- while you have reported error bars for number of opinion scores -- you have not incorporated the error for the limited number of speakers (i.e. each speaker has a bias as well, and taking the mean of 2 is not sufficient to be robust).
2.  The appendices I feel could use more work, or should be cut. At the moment (particularly appendix C), there are several strange text formatting issues, and some terms are explained while others are implied known. I recommend either (a) cut the derivation appendices down to just the gist, making it clear that full derivations are not provided, or (b) expand and refine the appendices to make it more understandable, clarify terms and where certain equations / identities are sourced from. At the moment, Appendices A-C are enough to whet the appetite on how you phrase each type under your unified framework, but not clear/expanded enough for most readers to actually follow all steps in the derivation. Put simply: the math derivation, while verbose, is not clear or easily understood by most readers, and the appendices are insufficient. They should either be partly abandoned to save space, or fleshed out.

**Questions:**

- The entire training procedure of Section 3.4 can be simplified to "We iteratively refine a noisy waveform $\mathbf{x}\_i$ by linearly
interpolating it toward $\mathbf{x}\_{predict} $ with weight $\frac{t\_i}{t\_{i-1}}$, where $t_i = t_{i-1} - \Delta t$ for some predefined number of steps $i=1...\textit{step}$ ". This seems to make a lot of intuitive sense why this helps and works. What I am not sure of, however, is how necessary it is to introduce all the theory around velocity fields and refinement? i.e. Algorithm 1 can be intuitively understood as a simple iterative linear interpolation of a running prediction toward the GAN model's prediction.
- $\mathcal{L}\_{Mel}$ is never defined, but is used in your final loss?? Further, $\mathcal{L}\_{Rec}$ is defined but never used?
- It is not immediately clear how equation (45) corresponds to the standard time-varying Poisson equation. Can this please be clarified? Similar to what is done for score based models in Appendix A. In fact, the first part of Appendix C is quite rough both on understanding and grammar, could likely do a rework.
- Could the code and samples be made more accessible? Currently, viewing the sample clips is quite cumbersome, and there is little documentation. The author's ideas are rather valuable, and making even the core code pieces and audio samples more accessible and easy to view and use, will go a long way in improving the value of this work.

**Details Of Ethics Concerns:**

None, no ethics concerns.

---

### Official Review · Reviewer_kPeF · 2023-11-07

**Soundness:** 2 fair
**Presentation:** 1 poor
**Contribution:** 3 good
**Rating:** 3
**Confidence:** 3

**Summary:**

This paper proposes a new model to perform the well-established task of predicting raw waveform from a mel-spectrogram. This type of model is known as a neural vocoder, and this task is typically known as speech synthesis or neural vocoding, and has many applications such as text-to-speech (via text-to-mel + vocoder), speech-to-speech (e.g., translation, voice conversion), video-to-speech, etc. Instead of estimating the solution to a  linear partial differential equation (PDE), as is done by diffusion models and Poisson flow generative models, the model learns to estimate the velocity field in a non-compressible passive fluid partial differential equation via a generative adversarial network (trained with a discriminator). The authors design a generator that upsamples the input spectrogram via convolutions, self-attention mechanisms, and upsampling blocks (attention upsampler) and refines its outputs using a U-Net model. The self-attention mechanisms in the attention upsample are carefully designed in a novel way based on a previous work focused on linear self-attention. The discriminator is symmetrical to the upsampler but has different hyperparameters and employs Discrete Wavelet Transform (DWT) instead of traditional downsampling to avoid aliasing and improve results. The objective is the sum of the LSGAN loss, mel-spec + DWT reconstruction loss, and a feature matching loss (typically use in GANs, such as HiFi-GAN). Many tricks are applied to make generation faster and more accurate, such as reparametrization techniques to make the refiner predict the original data, which reportedly enhances the speed of convergence, DropKey in the attention mechanisms to avoid overfitting, etc. The authors show that the method outperforms other recent models in MOS, PESQ, and is competitive on PESQ and inference speed (measured by RTF). The model also seems to generalize to unseen speakers favorably compared to other vocoders. The authors justify some of the unique design choices via a convincing ablation.

**Strengths:**

Overall it is clear that a lot of work has been put into this paper. The proposed method has strong mathematical motivations and the analysis of previous models through the lens of partial differential equations shows a good and somewhat unique understanding of previously proposed models. The methodology certainly does not lack novelty, featuring a novel prediction objective (PDE velocity estimation), and many novel components, down to the implementation of attention mechanisms in an efficient manner. It is clear that a lot of effort has gone into making the training and inference efficient, which is highly appreciated. The results seem to be on par with previous works, which is impressive since neural vocoding is a very competitive domain with carefully fine-tuned architectures and objectives.

**Weaknesses:**

Overall I find this paper extremely difficult to follow. The explanations in section 2 are generally clear, although are sometimes phrased awkwardly, making them difficult to understand at first glance. I am not very experienced with partial differential equations, but the formulations seem reasonable overall. More importantly however, the writing in the rest of the paper is very difficult to follow and sometimes too colloquial. Some examples include:
- "Actually we also adopted WGAN(Arjovsky et al., 2017) training method,but only found similar performance with LS-GAN." - The use of the word "actually" here is too informal in my opinion.
- "In addition to the objectives of GAN, we incorporate mel-spectrogram loss and Discrete Wavelet Transform (DWT) loss." - the authors should write "we incorporate *the* mel-spectrogram loss (...)", or something similar.
- "WaveGlow (Prenger et al., 2019), an ancient parallel flow-based model;" - The use of the word "ancient" seems inappropriate here
A lot of the content in the methodology is often confusing. For example, in section 3.2.1., the words "Generator" and "Attention Upsampler" seem to be confused, even though the attention upsampler is supposed to be part of the generator. Another instance is section 3.4, where I have a hard time understanding how this reparametrization works exactly. Algorithm 1 is helpful but not sufficient in my opinion. Figure 1 is incomplete, the inputs and outputs of each component should be detailed. Also, when there are multiple blocks (e..g, the purple upsampling block on the left), the number of blocks should be indicated, even if via a variable (i.e., "N blocks"). Finally, the evaluation is okay but not particularly impressive - it seems this method is extremely complicated and requires many tricks to work, but it is not convincingly better than the simpler HiFi-GAN for example, since it is slightly better in MOS but also slower. More speech metrics would also be appreciated (VisQOL, ESTOI, MCD, DNSMOS, etc.). Also, a comparison with BigVGAN (and perhaps BigVSAN) is necessary, since these are recent state-of-the-art vocoders. The discussion is also very short and lacks depth, which is unfortunate.

**Questions:**

- Section 3.2.3 "The discriminator is symmetric with the first stage generator, but has quite different super parameters
compared with the first stage generator". What is "super parameters" referring to here? Are you referring to hyperparameters? If not, I am unfamiliar with the term "super parameters".
- Section 3.2.1 " Subsequently, the output is the average of the outputs from both the generator and the refiner." How can the output be the average of the refiner and the generator, if the generator is composed of the upsampler and refiner? Do the authors mean to say that the output of the generator is the average from the outputs of the upsampler and refiner? If so, this should be rephrased. If not, I don't understand what is being said here.
- A more detailed description of how RTF is computed would be helpful since this can vary from paper to paper and the details can be very important (i.e., what is the length of the input sample, and how many samples is this metric averaged over?).

---

### Official Review · Reviewer_fHRe · 2023-11-07

**Soundness:** 3 good
**Presentation:** 3 good
**Contribution:** 2 fair
**Rating:** 5
**Confidence:** 5

**Summary:**

This paper proposes a new physical perspective to interpret the probabilistic modeling of generative models, using the partial differential equation of non-compressible fluids to describe the distribution transformation between the prior and the ground truth distribution. Based on this theorem, the paper introduces WaveFluid, a two-stage GAN that models the velocity field of distributions to generate high-fidelity audios from mel-spectrograms. This network introduces a generator and a refiner to enhance the quality of generation, and employs various performance optimization techniques. Experiments show that the proposed model achieves a good balance between generating speed and audio quality compared to previous GAN-based and score-based vocoders.

**Strengths:**

1.The physical perspective of using fluid equations to elucidate the modeling of probability distributions in generative models is highly innovative. It provides a theoretical foundation for better analyzing existing generative models and designing more optimal ones.

2.The article offers a comprehensive mathematical analysis of using fluid equations to explain diffusion models and Poisson flow models, providing strong evidence for this analytical approach.

3.The proposed model achieves a good balance between audio quality in generation and generation speed. The provided samples sound of good quality.

**Weaknesses:**

1.The experimental part is relatively limited. More advanced GAN-based vocoders, such as BigVGAN, may be compared as baselines. And additional metrics like MCD may be calculated. For results unseen speakers, evaluating speaker similarity with metrics like SMOS would also be valuable.

2.There's a lack of analysis concerning the model's size, including an analysis of the impact of different parameter counts on the model's performance and an analysis of the effect of varying parameter sizes of different baselines.

3.As described in the article, this method is only applicable to specific tasks with small information gaps between conditions and data (such as mel-conditioned audio generation) and is not suitable for general-purpose generation. This limitation restricts its broader application.

**Questions:**

Why are different datasets used for subjective and objective evaluations of unseen speaker results? Both results on LibriSpeech and Speech Command may be used to calculate MOS and objective metrics.

---

### Official Review · Reviewer_2TB9 · 2023-11-09

**Soundness:** 1 poor
**Presentation:** 2 fair
**Contribution:** 1 poor
**Rating:** 3
**Confidence:** 4

**Summary:**

This paper presents a novel approach and model architecture for mel-spectrogram conditioned speech synthesis, building upon previous research in denoising diffusion probabilistic models (DDPM) and Poisson flow generative models (PFGM). Similar to DDPM and PFGM, the approach incorporates a multi-step inference process, allowing users to flexibly adjust the number of sampling steps to strike a balance between efficiency and quality. In contrast to DDPM and PFGM, which estimate solutions to partial differential equations, the proposed method directly learns a velocity field and is trained using an adversarial loss, in contrast to the widely used reconstruction loss in diffusion models.

The proposed model architecture is comprised of three key components: a deterministic up-sampler, a probabilistic refiner (which handles the multi-step inference process), and a discriminator. They also employ certain techniques, such as the multi-period speech attention, and the utilization of Discrete Wavelet Transform (DWT) for non-destructive downsampling, to enhance both time and memory efficiency as well as overall performance.

The experimental results demonstrate that the proposed method achieves comparable performance to previous diffusion-based and GAN-based vocoders in both single-speaker and multi-speaker setups. Additionally, an ablation study is included to showcase the effectiveness of the proposed method and model architecture.

**Strengths:**

**Review of Diffusion and Poisson Flow Models**

This paper provides a well-structured review of generative models relevant to diffusion probabilistic models and Poisson flow models. These models are designed to estimate solutions to partial differential equations and involve a multi-step generation process. The appendix contains detailed explanations of model formulations and methods for solving the associated partial differential equations, offering a comprehensive overview of the technical background of diffusion-based models.

**Application of the Proposed Method to Mel-Spectrogram-conditioned Speech Synthesis**

The authors acknowledge that the proposed method may not be universally applicable to all generative tasks and is better suited for tasks with strong conditioning. Therefore, this paper primarily focuses on vocoding tasks. The mel-spectrogram input serves as a strong conditioning factor, constraining the possible outcomes of audio generation. Mel-conditioned speech synthesis therefore emerges as an ideal task for applying the proposed method.

**Weaknesses:**

The paper exhibits several areas where improvements can be made to enhance the clarity and presentation of its content. The following are my suggestions:

**Model Architecture**

One of the paper's main focal points is the enhanced model architecture. However, there is room for improvement in the presentation and experimentation related to this aspect. Some examples include:

- The inputs and outputs of individual model components lack clear explanations. For instance, in Figure 1, most of the inputs and outputs for each model component are not depicted. In Figure 1(b), it is shown that the Refiner module takes three inputs ($x, t, mel$), but there is an additional $x_{hint}$ input mentioned in Algorithm 1.
- The integration of the time parameter $t$ into the model is not well-documented. In Figure 6, it appears that $t$ is treated as a continuous scalar. However, it is common in prior research to use a time embedding to condition the generation process of diffusion-based models. The author should clarify how the time information is integrated into the generation process.
- Despite the paper emphasizing the improved model architecture, it lacks comprehensive results and ablation studies to validate the effectiveness of the proposed model architecture. For instance, there is no ablation study on the 2-stage generator-refiner architecture, nor is there a discussion regarding the benefits of training the generator and refiner jointly or separately. This lack of information weakens the paper's claims.
- The differences between MOS scores in Table 3 fall within the margin of error. Similarly, the STOI and PESQ metrics show limited differences. As a result, the ablation study does not provide strong support for the model architecture proposed in this paper.

In summary, the authors should enhance the presentation of their model architecture and conduct a more thorough ablation study to bolster their claims.

**Generation Process**

As mentioned by the authors, the proposed method exhibits stability issues when the number of steps increases, limiting its applicability to tasks with strong conditioning (which may also be considered as “easy tasks”). This constraint restricts the method's suitability for more diverse tasks, noisier acoustic environments, and a broader range of applications. These limitations negatively impact the method's appeal to potential readers.

**Experiments**

The presentation of results and experimental setups could also benefit from improvement. Here are some areas that could be enhanced:

- In the experiments involving unseen speakers (Table 2), different datasets are used for different metrics, creating inconsistencies in the experimental setups. The author should provide an explanation for these variations.
- The paper lacks sufficient information about the ablation study, such as the dataset used.

**Connection to Previous Work**

The last, but perhaps the most important part, is the connection between this work and the previous DDPM and PFGM models.

- It is arguable whether it makes sense to extend the concept of velocity field in the Poisson flow model to the GAN-based training in this work. In the Poisson flow model, the velocity field has a clear physical interpretation, while in this paper, the velocity field only divides the inference process of GAN into several steps.
- In diffusion models, when training for the reverse diffusion process, the model's input is obtained by adding noise to a real sample, with strong theoretical support for this approach. However, in the multi-step generation process proposed in this work, the input to the model is merely a model prediction. Therefore, prediction errors may accumulate across time steps, potentially reducing the training effectiveness.
- The connection between this work and previous diffusion-based vocoders appears weak. This affects the paper's organization, making it challenging for readers to relate diffusion-based methods in Section 2 to the methodology in Section 3. I suggest that the author positions this work as a GAN-based vocoder additionally conditioned on a time parameter $t$, which enables multi-step inference and improves overall performance, rather than framing it as an extension of diffusion-based methods.

**Questions:**

In Algorithm 1, it appears that the model performs $step$ forward passes but only one loss computation and backward pass at the end, which does not make sense to me. It would be great if the author can provide additional context and explanation of the training algorithm to clarify this aspect.